# Multicomponent Training Improves the Quality of Life of Older Adults at Risk of Frailty

**DOI:** 10.3390/healthcare11212844

**Published:** 2023-10-28

**Authors:** Ana Moradell, David Navarrete-Villanueva, Ángel Iván Fernández-García, Narcis Gusi, Jorge Pérez-Gómez, Marcela González-Gross, Ignacio Ara, José Antonio Casajús, Alba Gómez-Cabello, Germán Vicente-Rodríguez

**Affiliations:** 1GENUD (Growth, Exercise, NUtrition and Development) Research Group, Universidad de Zaragoza, 50009 Zaragoza, Spain; amoradell@unizar.es (A.M.); davidnavarrete1845@gmail.com (D.N.-V.); angelivanfg@unizar.es (Á.I.F.-G.); joseant@unizar.es (J.A.C.); agomez@unizar.es (A.G.-C.); 2Exercise and Health Spanish Research Net (EXERNET), 50009 Zaragoza, Spain; jorgepg100@unex.es (J.P.-G.); marcela.gonzalez.gross@upm.es (M.G.-G.); ignacio.ara@uclm.es (I.A.); 3Department of Physiatry and Nursing, Faculty of Health and Sport Science FCSD, University of Zaragoza, 50009 Zaragoza, Spain; 4Agrifood Research and Technology Centre of Aragón-IA2 (CITA-Universidad de Zaragoza), 50009 Zaragoza, Spain; 5Department of Physiatry and Nursing, Faculty of Health, University of Zaragoza, 50009 Zaragoza, Spain; 6HEME Research Group, University of Extremadura, 10003 Cáceres, Spain; ngusi@unex.es; 7ImFINE Research Group, Universidad Politécnica de Madrid, 28040 Madrid, Spain; 8Centro de Investigación Biomédica en Red de Fisiopatología de la Obesidad y Nutrición (CIBEROBN), Instituto de Salud Carlos III, 28040 Madrid, Spain; 9GENUD-Toledo Research Group, Universidad de Castilla-La Mancha, 45071 Toledo, Spain; 10Biomedical Research Networking Center on Frailty and Healthy Aging (CIBERFES), Instituto de Salud Carlos III, 28029 Madrid, Spain; 11Defense University Center, 50090 Zaragoza, Spain

**Keywords:** body composition, fatigue, exercise, fitness, nutrition

## Abstract

Achieving a high quality of life in older adults can be difficult if they have limited physical function. The aims of this study were to evaluate the relationship between baseline values and variations in body composition, fitness, and nutritional status on health-related quality of life (HRQoL) and to describe the effects of a 6-month multicomponent training (MCT) programme and a 4-month detraining period on HRQoL. A total of 106 participants with limited physical function were included in this study (age: 80.8 ± 5.9 years; 74 females) and were divided into two groups: control (CON) and intervention (TRAIN). HRQoL was measured using the EQ-5D-3L questionnaire and a visual analogue scale (EQ-VAS). Information on body composition, physical fitness, Mediterranean diet adherence, and nutritional status were obtained. Healthier baseline values for body composition, fitness and nutritional status were associated with better HRQoL (explaining 23.7–55.4%). The TRAIN group showed increased HRQoL during this 6-month MCT, showing group-by-time interaction (*p* < 0.05) and a deleterious effect of detraining. Changes in weight, arm strength, and aerobic capacity contributed to explaining 36% of the HRQoL changes obtained with MCT (all *p* < 0.05). This MCT improved HRQoL in older adults with limited physical function. However, HRQoL returned to baseline values after detraining. This study highlights the importance of performing ongoing programs in this population.

## 1. Introduction

Older people’s life expectancy has increased in recent decades [1]. With this longevity, the changes accompanying ageing lead to a physiological detriment that could be aggravated by lifestyle, disease, and other factors [1]. The increased life expectancy, in addition to changes in ageing, leads to a higher rate of morbidity and loss of functional capacity, with a subsequent decrease in quality of life (QoL) [2]. For this reason, society has the challenge of ensuring improvement in the older population’s QoL by preserving their functional capacity and maintaining their independence.

The term “health-related QoL” (HRQoL) refers to the self-perception focused on the impact of health on a person’s ability to lead a fulfilling life and has been used as an extended-used parameter to evaluate the efficiency of interventions in healthcare [3]. HRQoL has been widely associated with nutritional, social, physical, or cognitive factors, and in general terms with healthy lifestyle behaviours [4,5,6]. In particular, physical activity has been well studied as one of the most important modifiable factors related to HRQoL [7], which is probably why the majority of interventions have been focused on physical activity and physical exercise [8].

With regard to improving the HRQoL of older adults, a critical target group is the frail population. Frail older adults and those undergoing the frailty process (pre-frailty) have been described as having lower HRQoL than their robust counterparts [8]. Frailty is characterised as having poor functional capacity, being more susceptible to vulnerability, and having an increased risk of dependency [9]. Exercise and, more specifically, multicomponent training (MCT) programs have been demonstrated as the most efficient interventions to prevent and reverse this state of frailty [10]. However, to the best of our knowledge, there are no specific studies investigating the effects of MCT on HRQoL or on its dimensions (specifically measured using the EQ-5D-3L) in older adults at high risk of becoming frail. From a preventive and treatment point of view, this could help us to use exercise more accurately and efficiently. There is no knowledge of whether improvements obtained with this type of exercise are reversed after a detraining period, as happens with physical fitness [11]. Such information would be of great interest for designing efficient and sustainable strategies where the resting periods do not lead to loss of gains, thus maximising the benefits.

In addition, there is a lack of information on the relationship between changes obtained in HRQoL and changes in body composition, physical fitness, and nutritional status in older adults with limited physical function. Thus, all these aspects should be clarified in order to design more specific interventions that ensure an increase in HRQoL.

Therefore, the main objectives of this study were to (1) analyse the relationship between body composition, physical fitness, and nutritional status with HRQoL in older adults with limited physical function; (2) study the effect of a 6-month MCT programme and a 4-month detraining period on HRQoL and its dimensions in older adults with decreased functional capacity; and (3) investigate whether the changes in body composition, physical fitness, or nutritional status are associated with improvements in HRQoL during MCT.

## 2. Materials and Methods

### 2.1. Study Sample

This present study consists of an intervention with the main aim of improving physical function in older adults with decreased functional capacity. This work was carried out in the framework of the EXERNET-Elder 3.0 project between 2018 and 2020 in the city of Zaragoza. A sample size calculation was conducted with the programme G*Power (version 3.1.9.2, Heinrich-Heine-Universität Düsseldorf), obtaining a total sample of 28 people in the control and training groups. A two-tailored design with three time-point evaluations was specified. Values used were 0.05 for type I error, 0.02 for type II error (80% power), and an assumed mean effect of *f* = 0.025. However, a bigger sample was recruited with a total of 110 participants from four health centres and three nursing homes. Initially, four participants were excluded due to insufficient descriptive information. Inclusion criteria included the following: age > 65 years, no cancer or dementia, and limited physical function (score of 4–9 on the Short Physical Performance Battery, SPPB) [12]. Scores of 4–6 points on the SPPB indicate severely limited physical functional capacity, and scores of 7–9 indicate moderately limited physical functional capacity [13]. Participants were allocated by convenience to two groups: control (CON) and training (TRAIN). Non-randomisation was used in order to ensure assistance. Despite this fact, both groups were homogeneous with respect to the main study variables at baseline. Additionally, three health-related speeches were offered to both groups, with the main aim of improving control adherence. The topics were “functional capacity and frailty”, “nutritional recommendations for older adults”, and “physical exercise recommendations for older adults”. The study protocol and sample calculation are explained in detail elsewhere by Fernández-García et al. [14].

A structured interview questionnaire was used for each participant to collect general information and other health outcomes. In particular, the variables included in this article were as follows: smoking habits, mean of self-reported daily walking hours and sitting hours [15], adherence to a Mediterranean diet [16], Mini-Nutritional Assessment [17], and Mini-Mental Examination [18]. All measurements were assessed on two different days, in the same order, to ensure that all participants were under the same conditions. The participants were evaluated on three different occasions: at the beginning and end of the 6-month MCT and 4 months later after a detraining period.

All participants were informed about this research and given an informed consent form to sign before being included. This study was approved by the Hospital Universitario Fundación de Alcorcón (16/50; Alcorcón, Spain) and registered in clinicaltrials.gov (Reference number: NCT03831841). This study was developed in accordance with the Helsinki Declaration of 1964 (revised in Fortaleza, 2013) and the current Spanish legislation on human clinical research (Law 14/2007).

### 2.2. The Exernet-Elder 3.0 MCT Program

The intervention consisted of a 6-month programme with three weekly 1 h training sessions. They were supervised by exercise professionals with a maximum ratio of 12 participants per instructor. The number of participants per group ranged from 8 to 16. All sessions were organised with the following structure: 10 min warm-up, 35–40 min main exercises and 10–15 min cool-down. In particular, the weekly schedule consisted of three training sessions of 1 h each, consisting of two strength sessions on Mondays and Fridays and one endurance session on Wednesdays. During the strength session, older adults performed various exercises to improve the strength and power of their upper and lower limbs and trunk, as well as static balance and functional performance of the DLA. The endurance session was used to perform exercises to improve cardiorespiratory fitness levels, dynamic balance, coordination, and motor skills.

All exercises were adapted to the individual’s functional capacity. Progression and intensity were adjusted during the 6 months to ensure appropriate stimulation and to develop adaptations. Different phases were distinguished during the 6 months, familiarisation, accumulation, development, and maintenance. Borg’s rating of perceived exertion was asked at the end of the session. Additionally, some training variables were adapted to each participant according to their functional capacity at baseline. The complete training protocol according to the Consensus on Exercise Reporting Template [19] has been described in detail elsewhere [14].

### 2.3. HRQoL

The Spanish version of the EQ-5D-3L questionnaire was used to measure HRQoL [20]. The EQ-5D-3L includes five dimensions: mobility, self-care, usual activities, pain/discomfort, and anxiety/depression. Each dimension has three response options: no problems, some problems, and several problems. Upon combining the results from these five dimensions, a unique health state could be calculated to obtain an EQ-5D-3L index. The overall index was calculated using the Spanish time trade-off tariffs [20]. A score ranging from 0 (death) to 1 (fully functional QoL) estimates the HRQoL of the participants.

The questionnaire also includes a visual analogue scale (EQ-VAS). Participants had to indicate (on a scale of 0–1) what they thought their current health was at that moment (0, “worst imaginable health state”; 1, “best imaginable health state”).

The EQ-5D-3L has been used worldwide with available normative data for different ages [21].

### 2.4. Body Composition Measurements

Height was measured with a portable 2.10 m stadiometer (SECA, Hamburg, Germany) and waist and hip circumference were measured using Rosscraft Anthrotape (Rosscraft Innovations Inc., Vancouver, BC, Canada). Anthropometric measurements were taken following the standardised procedures of the Society for the Advancement of Kinanthropometry [22]. A standardised test using bioelectrical impedance (TANITA BC 418-MA, Tanita Corp., Tokyo, Japan) was performed to obtain weight, total fat mass, fat mass percentage, and fat-free mass. Once these measurements were taken, the body mass index (BMI) was calculated. To ensure standardisation of the measurements, all participants had to arrive early in the morning after fasting, micturate prior to measurement, and remove their shoes, socks, and heavy clothes. All measurements were performed by the same specialist researcher.

### 2.5. Physical Fitness Performance

The fitness battery comprised six physical fitness tests adapted from the “Senior Fitness Test Battery” [23] and a handgrip test. The tests were performed as follows: static balance (Flamingo Test) [24], strength of upper extremities (Arm Curl Test) [23], lower-body strength (Chair Stand Test) [23], flexibility of the upper extremities (Back Scratch Test) [23], flexibility of the lower extremities (Chair Sit-and-Reach Test) [23], agility/dynamic balance (8-Foot Up-and-Go Test) [23], maximum walking speed (30 m Walking Test) [24], aerobic capacity (6-Minute Walk Test) [23] and handgrip maximum strength (Takei TKK 5401, Tokyo, Japan).

### 2.6. Nutritional Status and Mediterranean Diet Adherence

The Mini-Nutritional Assessment questionnaire was used to collect information about nutritional status. From the 18 items asked, a total score of 0–31 was created, 31 being the best score and indicating good nutritional status [17]. According to the score obtained, the subjects were classified as “malnourished” (<17 points), “at risk of malnutrition” (17–23.5 points) or ”well nourished” (>23.5 points) [17].

Additionally, to obtain a global parameter for diet quality, the 14-item Mediterranean Diet Adherence questionnaire was administered. The total score for this questionnaire was 0–14 [16], with a higher score indicating better adherence. The following categories were created for adherence: low (0–5 points), moderate (6–9 points), and high (10–14 points).

### 2.7. Statistical Analysis

Statistical analyses were completed using the Statistical Package for the Social Sciences v. 20.0 for Windows (SPSS Inc., Chicago, IL, USA). The normality of the sampling distribution was studied using the Shapiro–Wilk test. Parametric tests were used due to the normality of the sample. Data were reported as mean ± standard deviation for continuous variables and as number (*n*) of participants and percentage (%) for categorical variables. The level of significance was established at *p* < 0.05.

Differences in descriptive characteristics at baseline were analysed via one-factor ANOVA. A possible ceiling effect was described, indicating the number of people with maximum punctuation in the EQ-5D-3L overall index at baseline.

To analyse possible predicting factors for HRQoL level, simple linear regression analyses adjusted by gender and age were performed between the independent variables (body composition, physical fitness, and nutritional status) and the dependent variables (EQ-5D-3L overall index and EQ-VAS). In particular, one linear regression analysis via the enter method was performed with each independent variable for each dependent variable.

Changes for the EQ-5D-3L overall index and the EQ-VAS were calculated by subtracting 6-month MCT values from baseline values, 4-month detraining values from 6-month MCT values, and 4-month detraining values from baseline values to obtain an overall value. A mixed-effect model was used to compare between and within changes during the 6-month MCT and the 4-month detraining period. To perform the analyses, the CON and TRAIN groups were established as a fixed factor, participants as a random factor, and baseline values, gender, and age as covariates. The models were calculated by considering maximum likelihood estimation. Bonferroni corrections were applied for post hoc pairwise comparisons. Moreover, several chi-squared tests were performed to find differences between groups (CON and TRAIN) in the percentage of people who improve, deteriorate, or maintain the punctuation in each of the dimensions of the EQ-5D-3L during training and detraining.

Then, to analyse whether changes in independent variables determine changes in HRQoL variables, additional multilinear regression analysis via the enter-method was performed with body composition and physical fitness changes as independent variables and with the EQ-5D-3L overall index and EQ-VAS as dependent variables. Variables were introduced using the enter method. All these changes were also calculated by subtracting 6-month MCT values from baseline values. Analyses were adjusted by gender and age and performed separately on the CON and TRAIN groups.

## 3. Results

### 3.1. Descriptive Characteristics of the Sample

Table 1 shows the descriptive characteristics of the whole sample and the CON (*n* = 55) and TRAIN (*n* = 51) groups separately at baseline.

The study sample did not show statistically significant differences in the SPPB, sex or age (*p* > 0.05). The TRAIN group spent more hours walking compared with the CON group. Higher values for weight, fat-free mass and handgrip strength were also observed in the TRAIN group (*p* < 0.05). No other statistically significant differences were observed for body composition, physical activity, or nutritional status variables between groups (all *p* > 0.05).

### 3.2. Baseline Relationships between the EQ-5D-3L Overall Index, the EQ-VAS and Body Composition, Physical Fitness, and Nutritional Variables

Appendix A shows the associations between body composition, physical fitness, and nutritional factors with the EQ-5D-3L overall index and the EQ-VAS adjusted by gender and age.

Except for fat-free mass, all body composition variables had a statistically significant association with the EQ-5D-3L overall index. Less weight, BMI, total and fat mass percentage, and waist and hip circumferences seem to predict better HRQoL, accounting for 36.5–51% of its variability (all *p* < 0.05).

For physical fitness parameters, better results in all tests (except for leg flexibility) seem to predict better scores in the EQ-5D-3L overall index. In particular, these variables account for 24–53% of the variability for arm flexibility and aerobic capacity, respectively (all *p* < 0.05).

Only the total score of nutritional status measured with the Mini Nutritional Assessment questionnaire seems to predict HRQoL, accounting for 30% of the model.

For the EQ-VAS, no body composition variables seem to predict the results. However, better results in agility, walking speed, and aerobic capacity seem related to a better score (all *p* < 0.05). Furthermore, higher adherence to a Mediterranean diet seems to be associated with better values for the EQ-VAS.

### 3.3. Effects of the MCT Programme and 4-Month Detraining Period on HRQoL

All participants who assisted in the evaluations and completed all data are included in the analyses. A possible ceiling effect in the EQ-5D-3L was observed in 10.5% of the CON group and 17% of the TRAIN group at baseline.

The effects of the training programme and the detraining period are presented in Table 2. Differences in participants included in each period were because they did not assist with the evaluations because of being on holiday (*n* = 20) or not fulfilling all the questionnaires (*n* = 10).

Significant differences were observed between changes in the CON and TRAIN groups during the 6-month MCT in both the EQ-5D-3L overall index (−0.014 ± 0.025 vs. 0.073 ± 0.02, respectively) and the EQ-VAS (−3.10 ± 3.00 vs. 10.60 ± 2.51, respectively) (all *p* < 0.05). The EQ-5D-3L overall index decreased in the CON group, whereas the EQ-VAS variables were significantly improved in the TRAIN group. Within-group changes were statistically significant only in the TRAIN group (all *p* < 0.05).

During the following 4 months of detraining, changes between groups were statistically significant only for the EQ-VAS (3.33 ± 3.41 for CON vs. −6.60 ± 2.53 for TRAIN). However, changes within groups showed statistically significant decreases in the TRAIN group for the EQ-5D-3L overall index (−0.065 ± 0.021) and the EQ-VAS (both *p* < 0.05); meanwhile, the CON group maintained these parameters through this period of time.

Overall, no significant changes were observed between or within groups from baseline to the end of the project 10 months later. However, it seems that the EQ-5D-3L overall index returned to baseline values, whereas the EQ-VAS only decreased by half of the improvements achieved.

Figure 1 and Figure 2 show the dimensions of the EQ-5D-3L overall index where changes and differences were observed between the CON and TRAIN groups during training and detraining. During training, significant differences were observed between the CON and TRAIN groups in the percentages of people who increased, maintained, or deteriorated in the dimension scores of anxiety or depression (*p* < 0.05), with a significant trend in pain or discomfort. During detraining, there were significant changes in the pain or discomfort dimension, where a greater number of CON participants seemed to improve, reporting less pain or discomfort (*p* < 0.05).

### 3.4. Influence of Changes in Body Composition and Physical Fitness Caused by Training and the Changes in HRQoL

Changes in arm strength during MCT in the TRAIN group contribute 32% to the change in the EQ-5D-3L overall index (*r*^2^ = 0.232, change in *r*^2^ = 0.084; *p* < 0.05). Moreover, changes in aerobic capacity contribute 36% (*r*^2^ = 0.205, change in *r*^2^ = 0.128; *p* < 0.05).

For the EQ-VAS variables, only changes in weight make an inverse contribution of 33% to the change in the EQ-5D-3L overall index (*r*^2^ = 0.113, change in *r*^2^ = 0.101; *p* < 0.05). All these results are included in Appendix A.

## 4. Discussion

The main findings of this study are as follows: (1) there is a direct relationship between healthier body composition, better physical performance, and higher HRQoL in older adults with limited physical function; (2) a 6-month MCT programme improves HRQoL in this population, whereas a 4-month detraining period reverses these benefits; and (3) improvements in weight, aerobic capacity and arm strength due to the training programme are directly associated with a 36% increase in HRQoL.

The relationships between HRQoL, body composition, physical fitness and nutritional parameters have been extensively explained in the literature on the older population. As expected, the results found are similar to previous studies when baseline values are analysed. For example, Olivares et al. have highlighted the importance of the Timed Up-and-Go Test and the 6-Minute Walk Test [25]. However, these were data from a younger sample (above 55 years). Moreover, in another sample of older adults, strength and walking speed showed a correlation with the EQ-VAS but not with flexibility, as we have observed [26]. The results suggest that independent of functional status, fitness levels are always positively associated with HRQoL. Likewise, adiposity parameters, BMI [5] and waist circumferences [27] present similar relationships with QoL. However, we did not find associations between muscle and HRQoL; in contrast, Verlaan et al. described a relationship when considering sarcopenic and non-sarcopenic older adults [28]. Nevertheless, sarcopenia not only depends on muscle mass but also includes strength in its definition [29]. Thus, more comparable methodologies should be used to confirm these results with different and more specific body composition measurements, such as dual X-ray absorptiometry. In addition, as found in larger sample studies, a Mediterranean diet and nutritional status are correlated positively with HRQoL, as they are strongly related to health parameters [6,30]. Concretely, nutritional status accounts for nearly 30% of the HRQoL which could be explained because it included a large range of parameters such as medication, functionality, pathologies, and body composition among others.

This study has described how the MCT programme improves the QoL of older people, using both the EQ-5D-3L overall index and the self-reported EQ-VAS. Previous studies analysing the effects of different exercise types have obtained contradictory results. A meta-analysis evaluating the effects of all types of exercise in the HRQoL of frail older adults did not find a significant effect of type of exercise on HRQoL, but they used the Short Form Health Survey of 36 items [31]. However, they ensured that studies with a positive effect on functional capacity report an effect on HRQoL, suggesting a relationship between both parameters [31], as in this present study. In this respect, a recent systematic review involving different types of exercise and questionnaires in a frail population established also this conclusion [32]. Furthermore, they did not find that a specific type of exercise was strongly related to improvement in HRQoL [32]. In addition, other authors showed that anxiety and depression were the dimensions more influenced by fitness levels [25], so it could be expected that the improvement in HRQoL is not only due to the beneficial effect on functional capacity. Therefore, as the EQ-5D-3L is required in the cost-effectiveness calculation [33], the present results show that the benefits obtained with this type of intervention would probably exceed the cost to healthcare institutions that this population would normally impose.

Global ageing is also a challenge from a social, political, and economic point of view, as the increase in the number of older adults may increase the pressure on healthcare systems and even the pressure on families, as the burden of caring for the elderly falls on them. Our study shows the positive effects of exercise but also focuses on the sustainability of the benefits achieved. The detraining analysis is, therefore, a novelty of this report. In fact, few studies focus their analysis on how the gains made during an exercise program are lost when the exercise program ceases and even fewer focus on quality of life. This is the first study evaluating its effect on health-reported quality of life after a multicomponent training program in people with limited physical function. In line with our results where TRAIN decrease in HRQoL, those Spanish habitual exercisers older adults participating in a two-week multicomponent program who stopped during 3 months in summer observed a similar decline [34]. Moreover, the reversibility of exercise effects on QoL has also been observed during the COVID-19 pandemic, in which there was a large population of older adults forced to stop their physical exercise activities [35]. These findings, along with our own, highlight that it does not seem to matter how long a programme or active lifestyle a person may have because when training or practice stops, the body’s functional, physical, and quality of life deterioration returns. It could be then hypothesised that people who are physically active and lose the ability to move or have their activity reduced, suffer a greater or more rapid deterioration in quality of life for various reasons, including physiological as well as mental aspects [36]. Declines in HRQoL in this period have been well described in physically active older adults [36,37]. Moreover, it seems that those with a decline in functional capacity are more susceptible to changing their HRQoL concretely in the depressive dimension [37]. The World Health Organisation (WHO) defines “healthy ageing” as the process of developing and maintaining functional capacity that enables well-being in older age [38]. And it puts the focus on development and maintenance, it seems clear that exercise develops and improves function, but if we want to fully meet the objective, it is necessary to maintain exercise programmes, in this case, MCT in the long term.

Finally, another important contribution of this study is the identification of specific improvements in parameters that are related to improvements in HRQoL. As mentioned above, several studies concluded that improvements in physical function and physical fitness are accompanied by improvements in HRQoL [31,32]. Concretely, this present study has identified arm strength, aerobic capacity, and weight loss as the most important contributors. A possible explanation could be their role in daily activities. For example, although leg strength is very important for independent living, higher aerobic capacity allows people to spend more time outside and be more active, which may improve QoL. Thus, despite the need for more research in this field, these parameters should be a priority when an intervention in this population is designed.

This study is not without limitations, even after attempting to control them in the methodological design. Firstly, randomisation of the sample was not possible because it is difficult to change older adults’ routines, and some participants refused to participate in the TRAIN group. Additionally, the randomisation of the sample was not performed for pragmatic and ethical reasons since not prescribing exercise to older adults may be considered unethical. Therefore, the sample was divided into CON and TRAIN groups according to the volunteers’ preferences/availability in order to maximise training attendance. Despite this fact, baseline characteristics were similar and may not be influencing the results. The EQ-5D-3L presents a ceiling effect in those participants who have good health status. However, it is a widely used questionnaire in health economic research and, on its own, easily describes the self-perception of HRQoL and can be extended easily to the older population. Furthermore, as presented here, there are few participants presenting this ceiling effect at baseline, which confers enough credibility to our positive results obtained with the exercise intervention. Although we tried to standardise all measurements to avoid possible bias, the bioelectrical impedance analysis has been criticised, and its accuracy is under discussion; future studies may use more precise methods. Therefore, as this study does not include a large sample, future studies with greater numbers of participants should be performed to establish sound conclusions. Among others, an important strength of this study is the in-depth description of the training protocol, as trial descriptions of exercise interventions are often suboptimal and leave practitioners or trainers unclear about the content of effective programmes [14].

## 5. Conclusions

In conclusion, healthier values for physical fitness, body composition, and nutritional parameters in older adults with limited physical function are associated with higher HRQoL. It has been shown that our MCT programme improved HRQoL in this population and that when exercise is stopped, this self-perception of health declines to basal values. In addition, it has been found that changes in HRQoL during the exercise intervention seem to be determined by improvement in arm strength and aerobic capacity and by weight loss. Thus, there is no doubt that an MCT programme is a useful type of exercise to improve QoL in people with limited physical function. This study highlights the importance of performing ongoing programmes in this population for sustainable and effective benefit. Furthermore, these physical fitness parameters and weight loss should determine how these exercise strategies are designed. Future research should be focused on evaluating the cost-effectiveness of these interventions.

## Figures and Tables

**Figure 1 healthcare-11-02844-f001:**
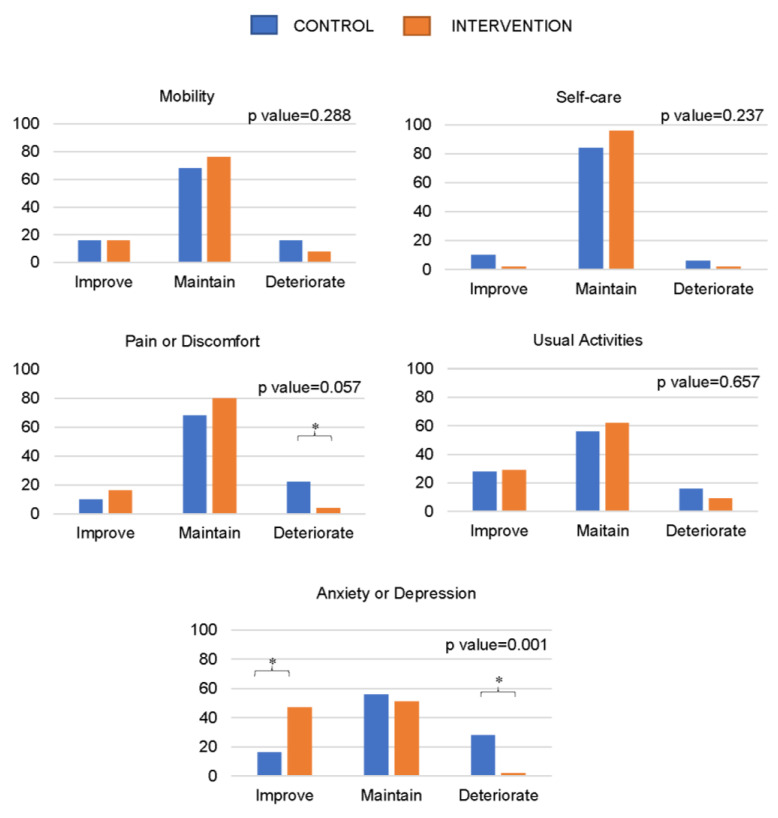
Percentage of people improving, maintaining, or deteriorating in different levels included in the EQ-5D-3L questionnaire during the intervention. * Significant differences between categories. *p*-value refers to the chi-squared test. All statistically significant differences were established at *p* < 0.05.

**Figure 2 healthcare-11-02844-f002:**
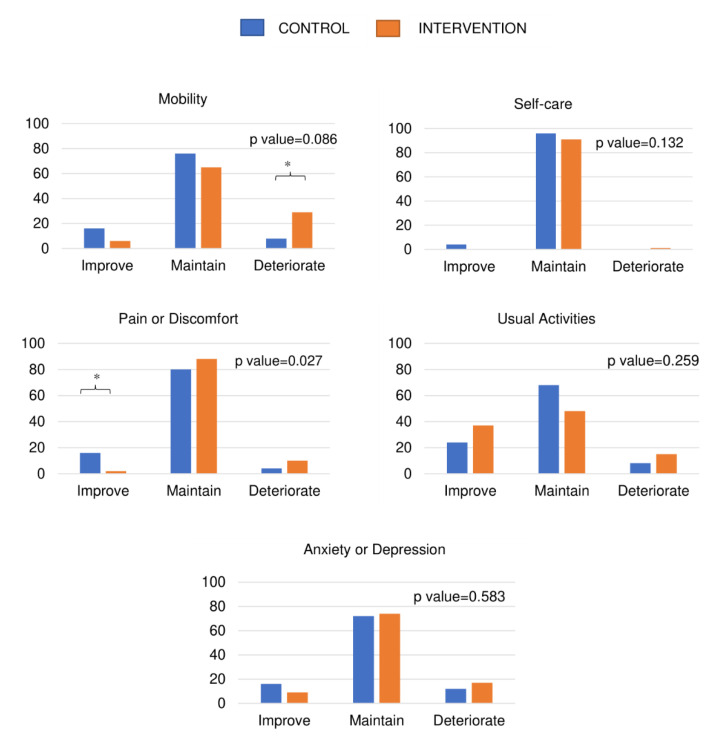
Percentage of people improving, maintaining, or deteriorating in different levels included in the EQ-5D-3L questionnaire, during the detraining period. * Significant differences between categories. *p*-value refers to the chi-squared test. All statistically significant differences were established at *p* < 0.05.

**Table 1 healthcare-11-02844-t001:** Descriptive characteristics of the sample at baseline.

Characteristic	Whole Sample	Control (*n* = 55)	Training (*n* = 51)	*p*-Value
Age (y)	80.8 ± 5.9	80.5 ± 5.7	81.2 ± 6.2	0.550
Sex				0.187
Male	32 (33)	14 (25)	18 (35)	
Female	74 (77)	41 (74)	33 (68)	
SPPB	7.5 ± 1.6	7.5 ± 1.6	7.5 ± 1.5	0.991
Severe limited functional capacity	30 (31)	16 (29)	14 (27)	0.512
Moderate limited functional capacity	76 (79)	39 (71)	37 (73)	
EQ-5D-3L	0.793 ± 0.186	0.773 ± 0.192	0.816 ± 0.178	0.236
Ceiling effect	15 (13.6)	6 (10.5)	9 (17.0)	0.240
EQ-VAS	62.9 ± 18.5	61.1 ± 19.1	64.8 ± 17.8	0.301
Walking hours per day	1.5 ± 1.2	1.13 ± 0.9	1.9 ± 1.4	0.001
Sitting hours per day	6.3 ± 2.9	6.3 ± 2.3	6.2 ± 3.1	0.860
Smoke				
Yes	5 (5)	4 (7)	1 (2)	0.206
No	101 (95)	51 (93)	50 (98)	
Cognitive Status	26.2 ± 3.1	26.1 ± 2.9	26.3 ± 3.0	0.713
MNA *	24.1 ± 3.6	23.7 ± 4.2	24.5 ± 2.9	0.302
At risk of malnutrition	34 (36)	18 (39)	16 (33)	0.356
Normal nutritional status	60 (64)	28(61)	32 (67)	
AMD	7.5 ± 1.9	7.5 ± 1.9	7.5 ± 1.9	0.913
Low adherence	12 (11.3)	8 (14.5)	4 (7.8)	0.522
Moderate adherence	77 (72.6)	37 (67.3)	40 (78.5)	
High adherence	17 (16.1)	10 (18.2)	7 (13.7)	
Body composition variables				
Weight (kg)	73.4 ± 14.8	69.7 ± 14.0	73.4 ± 14.8	0.015
BMI (kg/cm^2^)	29.8 ± 5.6	29.2 ± 6.0	30.3 ± 5.2	0.378
FM (kg)	27.6 ± 9.1	26.5 ± 9.5	28.6 ± 8.7	0.271
FFM (kg)	45.6 ± 9.1	43.2 ± 7.6	47.9 ± 6.7	0.014
FM%	37.1 ± 7.3	37.1 ± 8.0	37.1 ± 6.7	0.974
Waist circum. (cm)	94.4 ± 13.2	96.0 ± 12.0	94.4 ± 13.2	0.303
Hip circum. (cm)	104.6 ± 10.3	104.6 ± 11.9	104.6 ± 8.9	0.987
Physical fitness variables				
Balance (s)	6.8 ± 7.4	7.5 ± 8.8	6.0 ± 5.7	0.344
Arm Flexibility (cm)	−9.1 ± 9.8	−9.1 ± 11.5	−9.1 ± 8.5	0.979
Leg Flexibility(cm)	−12.4 ± 3.8	−12.7 ± 12.5	−15.0 ± 10.0	0.308
Leg Strength (rep)	10.2 ±.4	9.9 ± 3.3	9.7 ± 3.4	0.392
Arm Strength (rep)	12.4 ± 3.8	11.9 ± 4.0	12.9 ± 3.5	0.164
Agility (s)	9.8 ± 4.0	9.9 ± 4.1	9.7 ± 4.1	0.881
Walking speed (s)	26.9 ± 9.7	27.6 ± 9.4	26.1 ± 10.1	0.431
Aerobic capacity (m)	354.2 ± 106.4	337.4 ± 113.2	370.6 ± 97.6	0.117
Handgrip Strength (kg)	20.0 ± 8.1	18.0 ± 6.3	22.1 ± 9.2	0.010

SPPB: Short Physical Performance battery. EQ-VAS: EuroQol Visual Analogue Scale, EQ-5D-3L: EuroQol 5 Dimensions 3 Levels, BMI: Body Mass Index, FM: Fat Mass, FFM: Fat-Free Mass, FM%: Fat Mass Percentage, Circum: Circumference, MNA: Mini-Nutritional Assessment, ADM: Adherence to Mediterranean Diet, rep: repetitions. Categorical variables are presented as *n* (%). Significant *p*-values were set as <0.05. * Some participants did not complete the Mini-Nutritional Assessment questionnaire.

**Table 2 healthcare-11-02844-t002:** Changes during the 6-month multicomponent training and the 4-month detraining.

	6 Months Training			4 Months Detraining			Total 10 Months		
	Control (*n* = 31)	Train (*n* = 45)	*p*-Value	Control (*n* = 16)	Train (*n* = 37)	*p*-Value	Control (*n* = 28)	Train (*n* = 48)	*p*-Value
EQ-5D-3L	−0.014 ± 0.025	0.073 ± 0.021 *	0.009	−0.011 ± 0.029	−0.065 ± 0.021 *	0.146	−0.013 ± 0.031	−0.010 ± 0.023	0.965
EQ-VAS	−3.10 ± 3.00	10.60 ± 2.51 *	0.001	3.33 ± 3.41	−6.60 ± 2.53 *	0.022	−0.19 ± 4.5	5.72 ± 3.2	0.318

EQ-5D-3L: EuroQol 5 Dimensions 3 Levels (Total score), EQ-VAS: EuroQol Visual Analogue Scale. Statistical significance was established at <0.05. * Statistical significance within groups over time. *p*-value describes differences between groups.

## Data Availability

The data presented in this study are available on request from the corresponding author. The data are not publicly available due to privacy restrictions.

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
