# Peer review of "Multicomponent Training Improves the Quality of Life of Older Adults at Risk of Frailty"

_healthcare, 2023, doi:10.3390/healthcare11212844_

Round 1
Reviewer 1 Report
Comments and Suggestions for Authors
Thanks for the opportunity to review the manuscript, and I commend the authors for their valuable contribution. Here are my feedback and areas for improvement in the manuscript.
GxT p in the abstract, what does it stand for?
The Authors need to explain why Participants were allocated by convenience to two groups: control (CON) and training (TRAIN), what was the process and why there was no randomization to avoid bias?
What was the control group receiving?
Can the authors provide reference to the following statement "Anthropometric measurements were taken following the standardized procedures of the Society for the Advancement of Kinanthropometry" or simply mention the procedure followed during the anthropometric measurement
The statistical tests utelised are not clear enough, for example the analysis used to predict factors for HRQOL what type of linear regression (stepwise, enter method)used and why correcting for age since the age group of the participants are the same?
SPPB in table 1 is not introduced and multiple abbreviations are not introduced at the legend of table 1
Overall discussion is well-written
Conclusion is repeated twice.
Author Response
First of all, we would like to thank editor and reviewers for the time devoted for this constructive review, for they positive comments and for giving us the opportunity to resubmit the manuscript as a minor review. We have taken into account every single comment and here we attach a point by point response to all of them; accordingly we have also highlighted any changes made to the manuscript in yellow.
We believe that the manuscript has been improved accordingly and it is strong enough after this important review process.
Reviewer 2
Comment 1: GxT p in the abstract, what does it stand for?
Response 1: It refers to the group by time interaction. It has been explained instead.
Comment 2: The Authors need to explain why participants were allocated by convenience to two groups: control (CON) and training (TRAIN), what was the process and why there was no randomization to avoid bias?
Response 2: Thanks for the comment. The Non-randomization was done because there are some people who state during the recruitment, they were not going to train three days a week. It is necessary to know that they have a limited functional capacity and the fact of going three days a week would be hard for them. Moreover, as they express their repulsed, we decided to allocate by convenience in order to avoid drop-off to the train. Between those participants who did not mind participating (as they were recommended by their nurse or doctor), we create the training group till complete it in order they arrived and complete the recruitment. Although we are convinced that the ideal design is randomised, this decision has principles based on aspects of resource utilisation and ethics as we were also convinced about the benefits of the MCT it does not seem reasonable to include people in the programme who would probably not take advantage of it, further deteriorating the sample size and jeopardising the project, when others showed a predisposition to participate in the activity.
Comment 3: What was the control group receiving?
Response 3: During the project, all participants (control and intervention) received three one-hour educational talks related to healthy habits. The purpose of it was to reduce the possible drop-off caused by multiple evaluation periods, especially in the control group. The topics were “functional capacity and frailty”, “nutritional recommendations for older adults” and “physical exercise recommendations for older adults”. They were delivered by a certified nurse, nutritionist, and sport scientist respectively. We have extended this information in the main text.
Comment 4: Can the authors provide reference to the following statement "Anthropometric measurements were taken following the standardized procedures of the Society for the Advancement of Kinanthropometry" or simply mention the procedure followed during the anthropometric measurement.
Response 4: Thanks for the comment. The reference has been included in the text.
Comment 5: The statistical tests utilised are not clear enough, for example the analysis used to predict factors for HRQOL what type of linear regression (stepwise, enter method) used and why correcting for age since the age group of the participants are the same?
Response 5: Thaks for the comment. The reviewer is right. We have used the enter method as sex and age are well-studied to influence in body composition and fitness. Although the age group is the same for all participants, it could be an interaction effect in the variables. It is not strange to find difference in variables between participants in their sixties and their eighties. Thus, including age as a covariate helps control for potential age-related differences that might exist within the group.
Comment 6: SPPB in table 1 is not introduced and multiple abbreviations are not introduced at the legend of table 1.
Response 6: Thank you for this appreciation, all abbreviations have been now mentioned in the legend table.
Reviewer 2 Report
Comments and Suggestions for Authors
Comments on the Quality of English LanguageAuthor Response
First of all, we would like to thank editor and reviewers for the time devoted for this constructive review, for they positive comments and for giving us the opportunity to resubmit the manuscript as a minor review. We have taken into account every single comment and here we attach a point by point response to all of them; accordingly we have also highlighted any changes made to the manuscript in yellow.
We believe that the manuscript has been improved accordingly and it is strong enough after this important review process.
Comment 1: The authors describe the effects of a multicomponent (MCT) exercise program compared to a no-treatment control on an older, pre-frail sample. The program is only briefly described; more detail is needed to understand the results reported without searching for another paper the authors wrote.
Response 1: Thanks for the suggestion, we have added detailed information about the exercise program. If the reviewer considered there is more specific information that should be appeared let us know.
Comment 2: Of some concern, participants were not randomly assigned to treatment or control arms of the study, a serious weakness exemplified by the statistically significant difference in walking between groups at baseline.
Response 2: Thanks for this appreciation. The reviewer was right. The fact that the hours of walking are statistically significantly different is an important limitation not only when looking at comparisons between groups, but also when considering that are more likely to assist to the training sessions. The fact because this is a non-randomized study is to ensure the assistance of participants, as coming three days a week for 6 months could be a big effort for this pre-frail population. Although we are convinced that the ideal design is randomised, for resource efficiency and for ethical reasons as we were also convinced about the benefits of the MCT it does not seem reasonable to include people in the programme who would probably not take advantage of it, further deteriorating the sample size, when others showed a predisposition to participate in the activity. However; we believe that this difference is not of a great importance in the study of training changes.
In addition, objective data on accelerometry (not included here, but reported in another study of the project (Fernández-García ÁI, Moradell A, Navarrete-Villanueva D, et al. Effects of Multicomponent Training Followed by a Detraining Period on Frailty Level and Functional Capacity of Older Adults with or at Risk of Frailty: Results of 10-Month Quasi-Experimental Study. Int J Environ Res Public Health. 2022;19(19):12417. Published 2022 Sep 29. doi:10.3390/ijerph191912417) does not show differences between groups in physical activity intensity at baseline. Another relevant characteristic such as physical function is also homogeneous between the two groups.
Comment 3: It is well known that exercise can produce measurable improvement in physical function in older adults. Thus, the amount of discussion in the paper of the results of participation in the MCT program could be considerably decreased. The results of detraining, however, received far less attention in the paper but those results should have received far more attention because it was a more novel portion of the study and the results are useful.
Response 3: Thanks for the suggestion, we have tried to sum up the third paragraph of the discussion related to improvement and physical function during training. We agree with the reviewer that the topic of detraining is new and important, in fact there are few studies in this area and even fewer directly related to quality of life, however, we have made an effort to deepen and broaden the discussion in this section according to the reviewer's suggestion. you can see the changes and expansion shaded in yellow in the text.
Comment 4: Another interesting point is the finding that the MNA results accounted for 40% of the predictive model. This also needs further discussion, including more information on the role of nutrition education in the MCT program.
Response 4: The reviewer is right. We have tried to clarify further this information in the discussion. Moreover, we have corrected the sentence as MNA accounts a 30%, as it is shown in the table 2, but it still needed a further explanation. However, the nutritional status measured in the MNA goes further than nutritional education and dietary intake. This nutritional status considered gastrointestinal pathologies, low weight and BMI, poor physical function and medication, between others. That’s probably why it has a strong relationship with health-related quality of life, as it is a whole of different factors. Meanwhile, our educational speeches were about a practical and general approach in which dietary intake recommendations and common challenges in older adults were explained. Thus, they shouldn´t have a high impact on training group improvements.
Comment 5: Finally, there are occasional phrases that did not translate well; these need to be corrected.
Response 5: English has been revised to ensure there are now no English translation mistakes.